# Sahlep (*Dactylorhiza osmanica*): Phytochemical Analyses by LC-HRMS, Molecular Docking, Antioxidant Activity, and Enzyme Inhibition Profiles

**DOI:** 10.3390/molecules27206907

**Published:** 2022-10-14

**Authors:** Hatice Kiziltas, Ahmet Ceyhan Goren, Saleh H. Alwasel, İlhami Gulcin

**Affiliations:** 1Department of Pharmacy Services, Vocational School of Health Services, Van Yuzuncu Yil University, Van 65080, Turkey; 2Department Chemistry, Faculty of Sciences, Gebze Technical University, Kocaeli 41400, Turkey; 3Department of Zoology, College of Science, King Saud University, Riyadh 11362, Saudi Arabia; 4Department of Chemistry, Faculty of Science, Ataturk University, Erzurum 25240, Turkey

**Keywords:** sahlep, *Dactylorhiza osmanica*, antioxidant activity, phenolic content, enzyme inhibition, molecular docking

## Abstract

Studies have shown an inverse correlation among age-related illnesses like coronary heart disease and cancer and intake of fruit and vegetable. Given the probable health benefits of natural antioxidants from plants, research on them has increased. *Dactylorhiza osmanica* is consumed as a food and traditional medicine plant in some regions of Turkey, so evaluation of the biological ability of this species is important. In this study, the amount of phenolic content (LC-HRMS), antioxidant activities and enzyme inhibitory properties of an endemic plant, *D. osmanica*, were investigated. The antioxidant capacities of an ethanol extract of *D. osmanica* aerial parts (EDOA) and roots (EDOR) were evaluated with various antioxidant methods. Additionally, the enzyme inhibitory effects of EDOA and EDOR were examined against acetylcholinesterase (AChE), α-glycosidase, and α-amylase enzymes, which are associated with common and global Alzheimer’s disease and diabetes mellitus. The IC_50_ values of EDOA against the enzymes were found to be 1.809, 1.098, and 0.726 mg/mL, respectively; and the IC_50_ values of EDOR against the enzymes were found to be 2.466, 0.442, and 0.415 mg/mL, respectively. Additionally, LC-HRMS analyses revealed *p*-Coumaric acid as the most plentiful phenolic in both EDOA (541.49 mg/g) and EDOR (559.22 mg/g). Furthermore, the molecular docking interaction of *p*-coumaric acid, quercitrin, and vanillic acid, which are the most plentiful phenolic compounds in the extracts, with AChE, α-glucosidase, and α-amylase, were evaluated using AutoDock Vina software. The rich phenolic content and the effective antioxidant ability and enzyme inhibition potentials of EDOA and EDOR may support the plant’s widespread food and traditional medicinal uses.

## 1. Introduction

The *Orchidaceae* family is one of the richest flowering plant groups in the world, and there are 24 genera and 170 taxa in Turkey [1]. Orchids have a very high economic value. Although it is known especially as an ornamental plant, many orchid species are commonly used in the foods and pharmacy industry. The vanilla flavor obtained from orchids is frequently traded [2]. Orchid species are used for treatment of Alzheimer’s and Parkinson’s diseases (AD and PD), anxiety, depression, cancer, chest pain, tuberculosis, intestinal disorders, dysentery, diarrhea, cough, cold, anemia, and are also used as an aphrodisiac in adults [2,3,4]. It has been reported that these medicinal properties are due to secondary compounds such as polyphenolic compounds, ascorbic acid, indole alkaloids, and saponins [4]. It has also been determined that orchid roots have many compounds such as polyphenols and glucomannan, which have strong antioxidant properties [3]. The flour made from orchid roots, in Mediterranean countries and Turkey, is traditionally used as a drink called “salep” or “sahlep” [2,5]. Glucomannan polysaccharide, which is composed of mannose and glucose, is the main component in orchid flour [2]. In addition to hot drinks, sahlep is widely used in bakery foods, as an additive for ice cream, and in confectionery and pharmaceuticals [4]. In Turkey, sahlep is prepared from approximately 120 orchid taxa representing genera such as Dactylorhiza, Orchis, and Ophrys [2]. The genus Dactylorhiza, which belongs to the *Orchidaceae* family, occurs in Europe, the Mediterranean, and Asia, and it has been determined that there are 13 species in Turkey. *Dactylorhiza osmanica* (*D. osmanica*), which is one of the endemic *Orchidaceae* species that is the subject of our study, is used in Turkey for strengthening, treating wounds and abscesses, relieving mental fatigue, as an anti-inflammatory [1].

Oxidation is essential to fuel the biological processes of living organisms. However, it can cause uncontrolled production of oxygen-induced free radicals [6,7,8]. The balance between reactive oxygen species (ROS) formation and antioxidants is controlled by the antioxidant defense system [6]. However, exposure to UV light, smoking and other environmental pollutants, and cell metabolism disorders also increase the body’s free radical level, becoming a defect in the cell’s redox balance [9]. Increased ROS attacks membranes, enzymes, lipids, carbohydrates, proteins, and DNA, and causes various disorders and diseases [6]. There are studies confirming the role of free radicals and ROS in diseases such as cancer, atherosclerosis, brain dysfunction, and cardiovascular diseases [10]. Antioxidants, which consist of certain enzymes, vitamins such as E or β-carotene, and organic substances such as phenols and amines, can stop radical chain reactions and counter the harmful effects of oxidation in tissues [11]. Synthetic antioxidants are often used currently [9]. Recently, synthetic antioxidants have been suspected to have carcinogenic effects and therefore their use has been limited, so there is significant attention given to the discovery of natural antioxidants that can be used as substitutes [6,12].

Plants constitute an important source of active compounds and thus can show strong antioxidant effects. The fact that plants contain these active compounds that create physiological effects in the human body adds to their medicinal value [13]. Medicinal plants are recognized as a resource for the prevention and treatment of many diseases [14,15]. Plant phytochemicals attract the attention of researchers in the therapy of some diseases such as metabolic inflammation and cancer. Some phytochemicals are the subject of cancer therapy research. Many phenolics and flavonoids are frequently used in the design and development of drug processes. Studies have determined that phenolic compounds in plants can reduce or prevent oxidative damage from free radicals [16]. Therefore, the commercial importance of plants containing these secondary compounds is increasing [17,18].

In this study, the antioxidant capacities of evaporated ethyl alcohol extract of aerial parts and roots of *D. osmanica* were determined by several different in vitro antioxidant activity methods: Fe^3+^-TPTZ reducing capacity (FRAP); ferric ion (Fe^3+^) reducing capacity; cupric ion (Cu^2+^) reducing capacity (CUPRAC); DPPH and ABTS radical removing methods; and Fe^2+^ chelating activity. Furthermore, the amount of phenolic content determined by LC-HRMS analysis was also investigated. The enzyme inhibition ability of the extracts was determined against AChE, α-glycosidase, and α-amylase enzymes. Additionally, the molecular docking interaction of the most plentiful phenolics in the extracts with AChE, α-glycosidase enzymes and α-amylase, were determined using the statistical program of IBM SPSS Statistics 20.

## 2. Results

Numerous antioxidant activity assessments are used for preventing oxidation processes. One of the most important of these is the DPPH radical removal method. DPPH^•^ scavenging abilities of EDOA and EDOR and standards were investigated. The IC_50_ values for EDOA and EDOR and standard antioxidants were as follows: Ascorbic acid (16.12 ± 0.003, r^2^: 0.9566) > α-Tocopherol (23.10 ± 0.032, r^2^: 0.9825) > BHT (31.50 ± 0.011, r^2^: 0.9754) > EDOA (86.63 ± 0.010, r^2^: 0.9894) > EDOR (115.50 ± 0.011, r^2^: 0.9794) (Table 1 and Figure 1a) (*p* < 0.001). The lower IC_50_ suggests an effective radical scavenging effect [19]. The IC_50_ values of ABTS^•+^ scavenging for EDOA and EDOR and standard antioxidants were determined in following order: EDOA (10.19 ± 0.002, r^2^: 0.9819) > α-Tocopherol (15.400 ± 0.003, r^2^: 0.9866) > EDOR (19.80 ± 0.013, r^2^: 0.9564) > Ascorbic acid (23.10 ± 0.001, r^2^: 0.9998) > BHT (26.65 ± 0.008, r^2^: 0.9717) (Table 1 and Figure 1b).

Additionally, EDOA and EDOR have an efficient ABTS^•+^ scavenging activity. A statistically significant difference was found between DPPH^•^ and ABTS^•+^ scavenging activity results measured in a concentration-based way (10–30 μg/mL) (*p* < 0.001) (Table 1). In the literature, no publication was found in which ABTS^•+^ scavenging activity was measured in *D. osmanica* extracts. This study provides a reference. In a study conducted with *Dactylorhiza hatagirea* tuber extract, ABTS^•+^ scavenging activity was found to be 1.06 ± 0.008 mM/g [4]. In another study, ethanol extracts of *Dactylorhiza romana* roots were investigated and the ABTS test IC_50_ was found to be 0.64 ± 0.005 mg/mL [20]. For ABTS^•+^ scavenging activity, the IC_50_ amounts obtained from this study and studies of the different species mentioned above were found to be quite effective.

Metal chelating activity of EDOA and EDOR and standard antioxidant compounds was evaluated and the IC_50_s was determined (Table 1 and Figure 1b). EDOA was realized to have the most effective chelating activity (*p* < 0.001, Table 1). The IC_50_s for the metal chelating ability of extracts and standard compounds was determined in the following order: EDOA (5.63 ± 0.033, r^2^: 0.9294) > BHT (14.75 ± 0.056, r^2^: 0.9646) > EDOR (46.20 ± 0.015, r^2^: 0.9000) > Ascorbic acid (99.0 ± 0.036, r^2^: 0.9985) > α-Tocopherol (330.0 ± 0.017, r^2^: 0.9109) (Table 1 and Figure 1c). The results indicate that EDOA and EDOR have very strong metal chelating activity. The literature search revealed no publications about Fe^2+^ chelating activity measured in *D. osmanica* extracts, so this study is the first.

In the Fe^3+^ reducing ability assay, the concentrations of standards, EDOA, and EDOR were increased steadily. The reducing ability of EDOA, EDOR, and standard antioxidants (120 µg/mL) were as follows: Ascorbic acid (1.52 ± 0.028, r^2^: 0.9970) > BHT (1.27 ± 0.005, r^2^: 0.9880) > α-tocopherol (0.99 ± 0.007, r^2^: 0.9942) > EDOA (0.99 ± 0.003, r^2^: 0.9553) > EDOA (0.83 ± 0.031, r^2^: 0.9423) (Table 2 and Figure 2a).

The CUPRAC of EDOA, EDOR, and positive controls were measured depending on the concentration (10–30 μg/mL) and were as follows: BHT (1.56 ± 0.089, r^2^: 0.9978) > Ascorbic acid (1.07 ± 0.007, r^2^: 0.9722 > α-Tocopherol (0.79 ± 0.061, r^2^: 0.9986) > EDOR (0.72 ± 0.022, r^2^: 0.9707) > EDOA (0.67 ± 0.019, r^2^: 0.9747) (Table 2 and Figure 2b). EDOA and EDOR demonstrated effective CUPRAC reduction ability (*p* < 0.001) (Table 2). The FRAP results of EDOA, EDOR, and standard antioxidants (150 µg/mL) decreased in the following order: Ascorbic acid (1.62 ± 0.015, r^2^: 0.9930) > BHT (0.91 ± 0.006, r^2^: 0.9874) > α-tocopherol (0.76 ± 0.075, r^2^: 0.9867) > EDOA (0.52 ± 0.005, r^2^: 0.9722) > EDOR (0.45 ± 0.006, r^2^: 0.9673) (Table 2 and Figure 2c).

The content of total phenolic in EDOA and EDOR was found to be 12.73 ± 1.29 and 9.09 ± 0.64 μg GAE, respectively. Additionally, the total content of flavonoids in EDOA and EDOR was found to be 17.54 ± 1.85 and 3.28 ± 0.12 μg QE, respectively (Table 3). It shows that a positive correlation was found between total phenolic and flavonoids in EDOA, EDOR, and antioxidant activities. In one study, the methanol extract of aerial parts of *D. osmanica* was investigated and the total phenol content was found to be 20.6 ± 0.379 μg GAE [1]. 

In the present study, the phenolic quantity of EDOA and EDOR was evaluated by LC-HRMS analysis. For this purpose, thirty-three phenolic compounds were quantified and identified (Table 4 and Figure 3). However, *p*-coumaric acid (541.49 mg/kg) and vanillic acid (62.22 mg/kg) are the main polyphenols identified in EDOA; *p*-coumaric acid (541.49 mg/kg) and ascorbic acid (42.93 mg/kg) are the main polyphenols in 1 mg of EDOR. It was determined that fumaric acid and *p*-coumaric acid are the most abundant polyphenols in both EDOA and EDOR.

The IC_50_ values measured for α-glycosidase were 1.098 μg/mL (r^2^: 0.9545) for EDOA; 0.442 μg/mL (r^2^: 0.9498) for EDOR; and 22.80 μM for Acarbose (Table 5) [21]. In addition, the IC_50_ values measured for α-amylase were 0.726 μg/mL (r^2^: 0.9860) for EDOA; 0.415 μg/mL (r^2^: 0.9747) for EDOR; and 10.01 μM for Acarbose (Table 5) Additionally, EDOA and EDOR had an efficient inhibition profile against the α-amylase as a proteolytic enzyme, with IC_50_ values of 0.726 (r^2^: 0.9860) and 0.415 μg/mL (r^2^: 0.9747), respectively (Table 5). It was reported that Acarbose exhibited α-glycosidase enzyme with an IC_50_ value of 10.01 μg/mL [22].

In the current study, the results of AD-related cholinesterase inhibition were evaluated and the IC_50_ values for AChE were measured to be 1.809 μg/mL (r^2^: 0.9722) for EDOA; 2.466 μg/mL (r^2^: 0.9826) for EDOR; and 0.124 μM for tacrine (Table 5) [24].

*p*-Coumaric acid, quercitrin, and vanillic acid were detected to be the main phenolic acid of EDOA and EDOR. However, it was reported that *p*-coumaric acid has been shown to have no inhibition toward AChE [25]. The best binding-pose selection was performed for quercitrin and vanillic acid, the two main compounds of EDOA and EDOR, by placing them into the active site of the AChE. Additionally, the best binding-pose selection was performed for *p*-coumaric acid and vanillic acid to the active site of α-amylase and α-glycosidase as other target enzymes. Docking studies were followed by analysis of binding modes to understand inhibition mechanisms. According to docking scores, quercitrin showed the highest binding affinity with AChE and *p*-coumaric acid showed the highest binding affinity with α-amylase and α-glycosidase enzyme targets (Table 6). Quercitrin and vanillic acid were placed in the active site of the enzyme AChE (PDB code: 4EY7). Figure 4B represents 3D and 2D interactions of Quercitrin–AChE and the docking score was calculated as −8.8 kcal/mol (Table 6). It is shown that the hydroxyl groups of quercitrin are linked to the active site through H-bond interactions with Gln-291, Arg-296, Glu-292, and Trp-286 active-site amino acids.

*p*-Coumaric acid-α-glycosidase (5NN8) complex’s docking score was calculated to be -6.5 kcal/mol (Table 6). A conventional H-bond of *p*-coumaric acid with α-glycosidase Asp-404 residue and two π anion interactions with Asp-518 and Arg-600 residues are shown in Figure 5B.

The binding affinity of *p*-coumaric acid-α-amylase (2QV4) complex was calculated as −5.6 kcal/mol (Table 6). *p*-Coumaric acid showed an H-bond with Asp-300 and Gln-63 residues in the active site of the α-amylase. Additionally, *p*-coumaric acid-α-amylase complex showed π-π stacked interactions with Tyr-62 residue in the active site Figure 6B.

## 3. Discussion

It is important to choose the most appropriate method when determining the antioxidant capacity of plants. In the present study, Fe^3+^ reducing, Cu^2+^ reducing and Fe^3+^-TPTZ reducing effects, ABTS and DPPH radical removal methods, and Fe^2+^ binding ability were used to determine the antioxidant capacity of the extract [26]. The DPPH method is based on the DPPH^•^ scavenging percentage of antioxidants in the plant extract. On the other hand, ABTS assay is based on the percentage of antioxidants in the plant extract to scavenge ABTS^•+^ radicals [27]. Ferrozine is known to form complexes with Fe^2+^ ions. In the presence of chelating agents in the environment, the generation of the complex is disrupted and leads to a decrease in the red complex color. In this way, the estimation of color reduction allows for estimating the chelating ability of the chelator [28].

Ascorbic acid was found to be the compound with the most effective DPPH^•^ scavenging activity. EDOA and EDOR were found to have a free radical scavenging ability close to standard compounds. In one study, DPPH^•^ scavenging activity of methanol extract of *D. osmanica* aerial parts was investigated and the IC_50_ value was found to be 0.1838 ± 0.0015 mg/mL [1]. In another study, DPPH^•^ scavenging activity of aqueous ethyl alcohol extract (70%) of *Dactylorhiza maculata* was investigated and the IC_50_ value was found to be 217.89 ± 10.89 mg ascorbic acid [29]. In a study conducted in 2020, extracts prepared from *Dactylorhiza romana* plant roots using different solvents were investigated, and the DPPH test IC_50_ of ethanol extract was calculated to be 1.53 ± 0.004 mg/mL [20]. When all these results were interpreted, it was determined that EDOA and EDOR did not exhibit a very strong DPPH^•^ scavenging activity. According to the ABTS^•+^ scavenging activity method, a stable form of the radical is produced in the experiment and forms blue-green ABTS^•+^ by reacting with an antioxidant, and decolorization specifies the rate of ABTS^•+^ inhibition [30,31].

Additionally, the results indicate that EDOA and EDOR have very strong metal chelating activity. In the literature search, there were no publications about Fe^2+^ chelating activity measured in *D. osmanica* extracts, so this study is the first. The antioxidant profile of EDOA and EDOR, characterized by using the ferric ion (Fe^3+^) reduction and CUPRAC and FRAP assays, are shown in Table 2 and Figure 2. Reduction capacity is an important factor in determining whether a molecule has antioxidant activity [32]. The first method used was to reduce Fe^3+^ to Fe^2+^ in Fe[(CN)_6_]^3+^ solution, which is one of the common methods. The reaction system is based on the reduction of Fe^+3^ in potassium ferricyanide to Fe^2+^ with the addition of an antioxidant agent and the formation of the Prussian blue color at 700 nm [19]. According to the results, it was determined that EDOA and EDOR have strong ability to reduce ferric ions (Fe^3+^) (*p* < 0.001) (Table 2). However, this value was found to be lower than standard antioxidants. In the CUPRAC test, the absorbance measurement of the stable complex occurred between neocuproine and Cu^2+^ ions, observed at 450 nm. High absorbance values indicate high reducing ability [19]. The Cu^2+^ ion reducing ability of EDOA, EDOR, and positive controls are demonstrated in Table 2 and Figure 2b. The FRAP assay is based on measuring the power of a sample with antioxidant properties to reduce oxidant ferric iron to ferrous form [33]. According to the method, higher absorbance values represent the higher reduction ability of the Fe^3+^-TPTZ complex. Furthermore, EDOA and EDOR demonstrated effective FRAP ability (*p* < 0.001) (Table 2). In one study, a methanol extract of aerial parts of *D. osmanica* was investigated and the Fe^3+^-TPTZ reducing value was found to be 804 ± 8.6217 (μM TE/g) [1]. *Dactylorhiza chuhensis* ethanol extracts were examined in a study and the FRAP values of tubers and flowers were found to be 85.3 ± 8.6 and 511.6 ± 252 µmol Fe^2+^/g DW, respectively [2].

The plants exhibited effective antioxidant capacity due to their secondary metabolites, including a large spectrum of phenolic and flavonoids [34]. Phenolic compounds are among the plant’s main secondary metabolites. It was determined that a diet rich in phenolic compounds has protective effects against cancer and cardiovascular diseases. Phenolic and flavonoids have many biological effects including anticancer, antibacterial, antiallergic, anti-inflammatory, and free radical scavenger [35]. Flavonoids form an important chemical class of secondary compounds in plants. Numerous phenolic hydroxyl groups attached to the ring structures of flavonoids give them antioxidant ability [36,37]. Owing to their strong free radical scavenging properties, flavonoids show antioxidant activities such as metal chelation and reduction [38]. In another study, total phenolic and flavonoid quantities in *Dactylorhiza hatagirea* tuber extract were found to be 11.42 ± 0.48 mg GAE/g and 11.46 ± 0.28 mg QE/g, respectively [4]. *Dactylorhiza chuhensis* ethanol extracts were examined in a study and the total phenolics of tubers and flowers were found to be 13.9 ± 0.6 and 44.2 ± 2.0 mg GAE/g DW, respectively [2]. In another study, the ethanol extract of *Dactylorhiza romana* plant roots was investigated and total phenolic and flavonoid contents were determined as 24.91 ± 0.95 mg GAE/g and 3.58 ± 0.08 QE/g, respectively [20]. The results obtained in these previous studies were found to be sometimes higher and sometimes lower than our results. The reason for this is thought to be due to differences in ecological and soil structure of the region where the plant is grown, analysis methods, solvents used, and extraction conditions.

It is known that antioxidant compounds including phenolics, flavonoids, and phenolic acids have a wide variety of pharmacological effects such as anti-inflammatory, anticarcinogenic and antiatherosclerotic activity [39]. Phenolic compounds, which have many beneficial effects on human health, are found in plants, vegetables, fruits, and cereals [40,41]. *p*-Coumaric acid is a hydroxyl derivative of cinnamic acid, and *p*-coumaric acid is one of its most abundant isomers in nature [42]. *p*-Coumaric acid is a natural phenolic acid found in many edible plants and exhibits various biological effectiveness as an antioxidant, antimicrobial, anti-inflammatory, and analgesic. It has also been determined to act as a tyrosinase inhibitor [42,43,44,45]. Few studies have been found to determine the phenolic content of *D. osmanica*. Only in 2018, a study was conducted with *D. osmanica*, which was collected from a different part of Turkey, and according to this, the most plentiful phenolics were determined by HPLC measurement to be syringaldehyde, *p*-coumaric acid, ferulic acid, synaptic acid, and benzoic acid [1]. The results of this research were found to be in agreement with the present study.

Diabetes mellitus (DM) is a metabolic disease caused by a disorder in insulin secretion. It causes chronic hyperglycemia and irregularity in carbohydrate, fat, and protein metabolism [46,47]. In the treatment of DM, compounds that inhibit the enzymes involved in carbohydrate absorption and metabolism are used, especially for the inhibition of pancreatic α-amylase and α-glucosidase enzymes, which are among the main enzymes involved in the intestinal absorption of glucose and play a key role in treatment. Inhibition of these enzymes reduces the absorption of sugars from the intestine and provides regulation of postprandial blood glucose level in Type 2-DM (T2DM) patients [48,49]. However, these drugs used for treatment have side effects. For this reason, natural compounds obtained from medicinal plants are being investigated for the treatment of T2DM, as they create better glycemic control and show fewer side effects [50]. Orchid species are among the plants widely used in traditional medicine due to their medicinal properties [48]. The determination of the inhibition of antidiabetic enzymes, α-glycosidase and α-amylase, was conducted for determination of antidiabetic capability of *D. osmanica*.

According to the results obtained, it was shown that ethanolic extracts of *D. osmanica* effectively inhibited α-amylase and α-glycosidase activities. These inhibitory effects were compared with Acarbose. In particular, EDOR had a very high affinity for α-amylase and α glycosidase. In the literature search, no data were found on the inhibitory properties of *D. osmanica* for α-glycosidase and α-amylase enzymes. However, there are studies carried out in different *Dactylorhiza* species. In a study conducted in 2020, the extracts prepared from the roots of *D. romana* with different solvents were investigated and the α-glucosidase and α-amylase inhibition IC_50_ values of the ethanol extract was determined to be 4.368 ± 0.053 and 76.554 ± 0.303 mmol/g, respectively [20]. In another study, α-glycosidase and α-amylase inhibition were investigated and the IC_50_ values of the methanol extract of *Dactylorhiza hatagireas* leaves were investigated and found to be 199.8 ± 4.7 and 210.28 ± 5.4 µg/mL, respectively [48]. In another study, α-glycosidase and α-amylase percentage inhibition values of *D. hatagirea* tuber extract were determined to be 46.80% and 27.97%, respectively [4]. The results of this study were consistent with the results of previous studies on the inhibition ability of different orchid species on α-amylase and α-glucosidase activities.

Alzheimer’s disease (AD) presents with memory loss and other behavioral abnormalities. In the treatment of AD, one of the most important methods is to control the level of acetylcholine by blocking the breakdown of acetylcholinesterase inhibitors [51]. AChE is an important enzyme that hydrolyzes neurotransmitter acetylcholine at cholinergic synapses in the central nervous system and peripheral nervous system [52]. Several compounds such as donepezil, galantamine, tacrine, and rivastigmine are used as AChE inhibitors in the treatment [53]. However, current AChE inhibitors have side effects and are only used to treat mild to moderate symptoms [52]. Medicinal plants are rich in different bioactive compounds including flavonoids and alkaloids, which are can be used in the treatment of some diseases including AD. Therefore, there is increasing interest in studies to obtain new drugs from plant extracts or compounds of plant origin [51,53]. It was determined that EDOA effectively inhibited AChE enzyme. In the literature search, no data were found on the inhibitory properties of *D. osmanica* for AChE enzyme. This work constitutes an initial reference for this. However, there are some studies carried out in several *Dactylorhiza* species. In a study, the extracts prepared from the roots of *Dactylorhiza iberica* and different solvents were investigated and AChE inhibition value of the methanol extract determined as 28.9% [51,54].

In this research study, data on the phytochemical bioactivity and properties, phenolic and flavonoid contents, antioxidant capacity, and enzyme inhibition potential of an endemic plant, *D. osmanica*, are presented for the first time. It was determined that *D. osmanica* contains a good level of phenolic and flavonoid contents and exhibits antioxidant activity close to standard compounds. In addition, it was also determined that there was a statistically significant difference between the averages of the groups. As a result of LC-HRMS analysis, it was determined that the main phenolic compounds of *D. osmanica* extracts are very rich in *p*-coumaric acid, vanillic acid, and quercitrin. Additionally, possible inhibition of *D. osmanica* extracts against α-glucosidase, α-amylase, and AChE enzymes was measured and it was determined that both EDOA and EDOR showed effective enzyme inhibition according to the results, which were also supported by molecular docking studies. The results obtained from this study with *D. osmanica* may provide a basis for studies to obtain secondary compounds of the species in pure form.

## 4. Materials and Methods

### 4.1. Chemicals

Chemicals for antioxidant ability were ordered from Sigma-Aldrich GmbH (Steinheim, Germany): α-tocopherol, neocuproine (2,9-dimethyl-1,10-phenanthroline), BHT (butylated hydroxytoluene), DPPH (1,1-diphenyl-2-picrylhydrazyl), ascorbic acid, ABTS (2,2-Azino-bis(3-ethylbenzothiazoline-6-sulfonic acid), Ferrozine (3-(2-pyridyl)-5,6-bis (4-phenyl-sulfonic acid)-1,2,4-triazine), and trichloroacetic acid (TCA).

Ascorbic acid (≥99%), chlorogenic acid (≥95%), fumaric acid (≥99%), caffeic acid (≥98%), vanillic acid (≥97%), naringin (≥90%), rutin (≥94%), syringic acid (≥95%), rosmarinic acid (≥96%), *p*-coumaric acid (≥98%), quercetin (≥95%), salicylic acid (≥98%), naringenin (≥95%), luteolin (95%), emodin (90%), and chrysin (≥96%) were purchased from Sigma-Aldrich. Orientin (>97%), (+)-*trans*-taxifolin (>97%), luteolin 7-glucoside (>97%), hyperoside (>97%), quercitrin (>97%), apigenin (>97%), hispidulin (>97%), acacetin (>97%), and hederagenin (>97%) were purchased from TRC Canada. Verbascoside (86.31%) was obtained from HWI Analytik GMBH. Luteolin-7-rutinoside (>97%) was purchased from Carbosynth Limited. Hesperidin (≥98%) was purchased from J&K. Dihydrokaempferol (>97%), isosakuranetin (>97%), and penduletin (>97%) was purchased from Phytolab. Apigenin 7-glucoside (>97%) was purchased from EDQM CS. Myricetin (>95%) was purchased from Carl Roth GmbH+Co. Nepetin (98%) was purchased from Supelco. Caffeic acid phenethyl ester (≥97%) was purchased from European Pharmacopoeia.

### 4.2. Plant Materials

Sahlep (*Dactylorhiza osmanica* var. osmanica (Klinge) P.F Hunt et Summerh) was collected from the Gevas district of Van province, south of Pınarbaşı, during June 2019 (Location: 38°16′42.1′′ N and 43°03′49.8′′ E). It was identified by Dr. Süleyman Mesut Pinar, Van Yüzüncü Yıl University, Faculty of Science, Biology Department. The voucher specimen (voucher code: MP 16425) is deposited at the Herbarium, Biology Department (VANF), Van Yüzüncü Yıl University, Faculty of Science, Van, Turkey.

### 4.3. Lyophilized Water Extract

The extraction procedures were performed according to the procedures previously described [55,56]. For the preparation of the ethanol extracts of *D. osmanica* aerial parts (EDOA) and roots (EDOR), the aerial parts and roots (each 25 g) of the shade-dried *D. osmanica* were first pulverized in a grinder. Then, the ground plant materials were soaked separately with 0.5 L of ethanol. The ethanol was evaporated (Heidolph Hei-VAP HL, Schwabach, Germany) and stored at −20 °C [57].

### 4.4. Radical Scavenging Methods

For evaluating the DPPH radical removing effect of the EDOA and EDOR, extracts and standards were prepared at different concentrations (10–30 μg/mL) and 1 mL of DPPH radicals (0.1 mM) was added to each sample tube. After 30 min of incubation, absorbance was recorded at 517 nm, as described previously [58]. To determine the ABTS radical scavenging effects of EDOA and EDOR, the method in a prior study was used [58]. For determination of ABTS^+•^ scavenging effects of EDOA and EDOR, the previously given method was used [59,60,61]. First, 2.45 mM persulfate solution was added to 2 mM ABTS solution to generate ABTS radicals. The absorbance of the ABTS^•+^ radical as control sample containing a 0.1 M phosphate buffer (pH 7.4) was adjusted to 0.750 ± 0.025 nm at 734 nm. Then, one mL of ABTS^•+^ solution was added to different EDOA and EDOR concentrations and after 30 min incubation absorbance were recorded at 734 nm [17,62]. Metal chelating ability was measured by inhibiting the formation of Fe^2+^–Ferrozine complex after treatment of test material with Fe^2+^ [63] with minor modification [64]. Fe^2+^-chelating effect was determined by the absorbance of the Fe^2+^–Ferrozine complex at 562 nm [65]. To summarize, different concentrations of EDOA and EDOR in 0.5 mL ethanol were transferred to a 0.1 mL of FeCl_2_ (0.6 mM). The reaction was started by adding 0.4 mL of Ferrozine (5 mM), which prepared in ethanol. The absorbance was measured spectrophotometrically at 562 nm [66].

### 4.5. Reducing Activity Methods

The Fe^3+^ reducing effects of EDOA and EDOR were measured depending on different concentrations (10–30 μg/mL). According to this method, the reducing capacity of an active molecule can be directly measured by reduction of Fe[(CN)_6_]_3_ to Fe[(CN)_6_]_2_ [67]. As a result, the Perl–Prussian blue complex, which exhibits absorbance at 700 nm, leads to the formation of Fe_4_[Fe(CN^−^)_6_]^3^ [68]. To determine the CUPRAC of EDOA and EDOR, a previous method with some changes was applied [69]. The FRAP method is based on the reduction of the TPTZ-Fe^3+^ complex [70].

### 4.6. Total Phenolic and Flavonoid Concentration

The quantity of phenolics in EDOA and EDOR was performed as described in previous studies [71,72]. The total amount of flavonoids found in EDOA and EDOR was determined as described before [73] and as given in a prior study [74].

### 4.7. Enzyme Inhibition Assay

The AChE enzyme inhibition properties of the extracts were determined according to a prior study [75]. The α-amylase and α-glycosidase inhibition effects of both extracts were estimated according to a method from previous studies [76,77]. The IC_50_ value is defined as the concentration of antioxidant compound causing 50% enzyme inhibition and was obtained from activity (%) against compound concentrations [78,79].

### 4.8. LC-HRMS Analysis

#### 4.8.1. Preparation of Samples and Conditions for LC-HRMS Analysis

The phenolic contents in EDOA and EDOR were determined by LC-HRMS analysis [80,81]. LC-HRMS analyses were performed on a Thermo ORBITRAP Q-EXACTIVE mass spectrometer (Bremen, Germany) equipped with a Troyasil (Istanbul Turkiye) C18 column (150 × 3 mm i.e., 3 µm particle size) for measurements. The mobile phases A and B were composed of 1% formic acid–water and 1% formic acid–methanol, respectively. The gradient program was 0–1.00 min 50% A and 50% B, 1.01–6.00 min 100% B, and finally 6.01–10 min 50% A and 50% B [82].

#### 4.8.2. LC-HRMS Procedure and Optimization of HPLC Methods

The final mobile phase included an acidified methyl alcohol and water gradient by the HPLC method [83]. The identification of the phenolics was made by comparing the retention times of the standard phenolics (in the purity range 95–99%; see section chemicals) and HRMS data of ILMER in Bezmialem Vakıf University. Dihydrocapsaicin (95%, purity) was used as the internal standard (IS) for LC-HRMS for reducing repeatability caused by external effects such as ionization repeatability in mass spectrometry measurements; 0.1 g/L dihydrocapsaicin (97%, Sigma-Aldrich) solution was used as the IS. The linear range of the standard solutions is given as mg/kg in Table 7. The mass parameters related to target compounds are summarized in Table 4 [82,83,84].

#### 4.8.3. Method Validation

Validation of the LC-HRMS method was performed by using analytical standards of corresponding compounds (see Section 4.1) as the target ions (Table 7). Considering the EURACHEM/CITAC Guide [84] and purpose of the method, the validation parameters were selected as linearity, recovery, repeatability, LOD, and LOQ for the applied method. The limit of detection (LOD) of the method for each compound was determined according to the following equation: LOD or LOQ = κSDa/b, where 3 is used for LOQ and κ = 3 for LOD; SDa represents the standard deviation of the intercept and b represents the slope (Table 7) [83].

### 4.9. Molecular Docking Studies 

The 3D version of compound chemical structures was downloaded from pubChem [84]. The 3D X-ray crystal structures of acetylcholinesterase (PDB ID: 4EY7) [85], α-glycosidase (PDB ID: 5NN8) [86] and α-amylase (PDB ID: 2QV4) [87] were downloaded from the “Protein Data Bank” website, with resolutions 2.35, 2.45, and 1.97 Å, respectively [88]. The structures of these enzymes were optimized in AutoDockTools 1.5.7 [89]. The most stable conformations and structure optimization of ligands were determined with AutoDockTools; the PDBQT file of the ligands was then prepared. The optimized enzyme and ligand structures were loaded into AutoDockTools and the same program was used for docking. The best scores of docking energy and binding interactions were analyzed with BIOVIA Discovery Studio.

### 4.10. Statistical Analysis 

Statistical analyses employed the unpaired Student’s *t*-test by using the statistical program of IBM SPSS Statistics 20. The results obtained were recorded as means with their standard deviation (SD); *p* < 0.05 was established as the minimum significance level.

## 5. Conclusions

In conclusion, in this study, EDOA and EDOR demonstrated effective antioxidant ability when compared to the standards, including BHA, BHT, α-Tocopherol, and Trolox. Additionally, EDOA and EDOR showed a value close to the standard compounds in all antioxidant activity tests. The ABTS^•+^ scavenging test showed better results than standard compounds. Both extracts possessed a wide spectrum of biological activities and can neutralize ROS and free radicals. EDOA and EDOR can be used to prevent or delay the formation of lipid autoxidation. Additionally, EDOA and EDOR were tested against enzymes such as acetylcholinesterase, butyrylcholinesterase, and α-glycosidase, which are associated with common diseases such as diabetes, Alzheimer’s disease, and glaucoma. Finally, the results indicated that EDOA and EDOR have some biological effects, including anticholinergic and antidiabetic effects. Thus, EDOA and EDOR may provide beneficial outcomes for treatment of diseases, following approval by further clinical and in vivo studies.

## Figures and Tables

**Figure 1 molecules-27-06907-f001:**
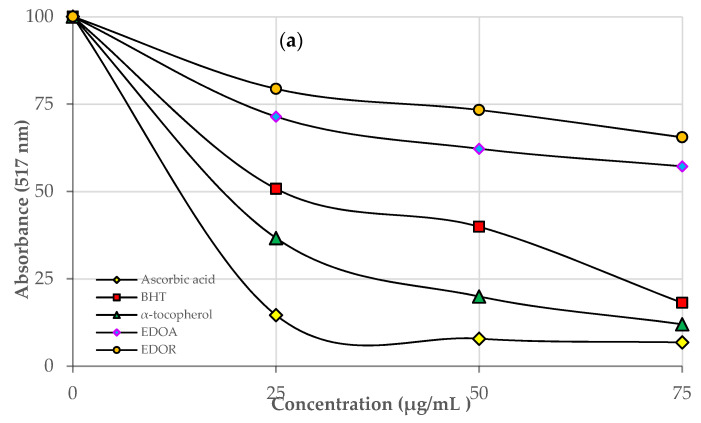
Radical scavenging ability of ethanol extracts of aerial parts (EDOA) and roots (EDOR) of sahlep (*D. osmanica*) and standards: (**a**) DPPH assay; (**b**) ABTS scavenging; (**c**) Fe2+ chelating.

**Figure 2 molecules-27-06907-f002:**
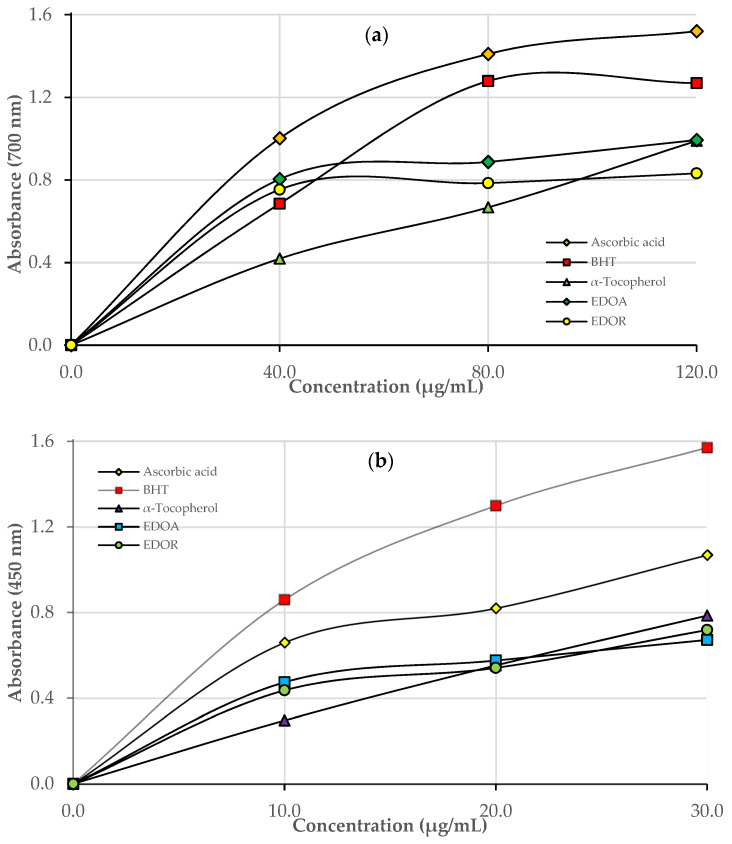
The reducing abilities of ethanol extract of aerial parts (EDOA) and roots (EDOR) of sahlep (*D. osmanica*) and standards: (**a**) Fe^3+^ reducing ability; (**b**) Cu^2+^ reducing ability; (**c**) FRAP reducing ability.

**Figure 3 molecules-27-06907-f003:**
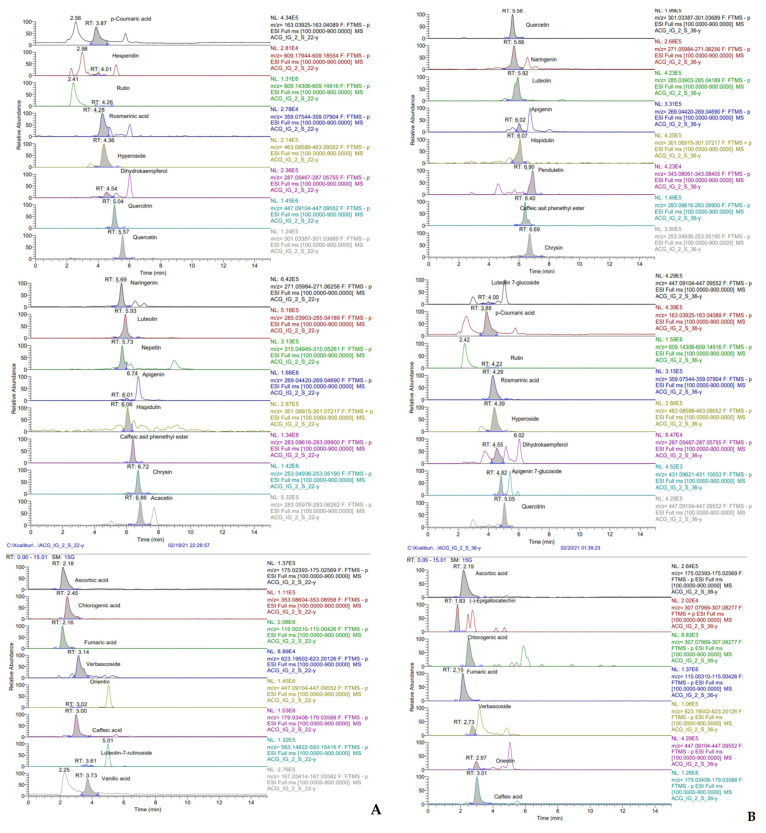
LC-MS/MS chromatograms of EDOA (**A**) and EDOR (**B**) from sahlep (*D. osmanica*) (EDOA: ethanol extract of *D. osmanica* aerial parts, EDOR: ethanol extract of *D. osmanica* roots).

**Figure 4 molecules-27-06907-f004:**
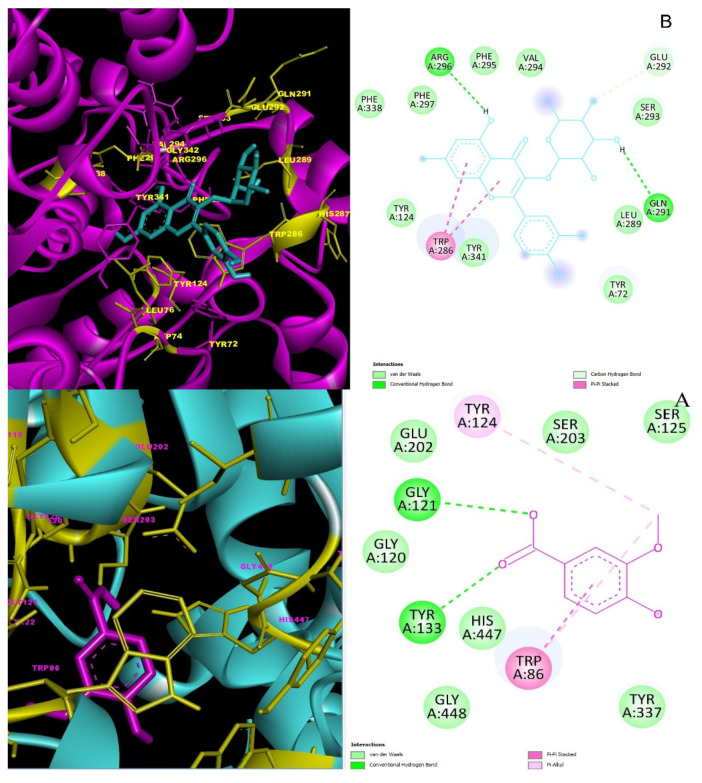
The 2D and 3D interaction profiles and best interaction poses of major phenolic compounds placed into the AChE (4EY7) by docking study: (**A**) Vanillic acid–AChE; (**B**) Quercitrin–AChE.

**Figure 5 molecules-27-06907-f005:**
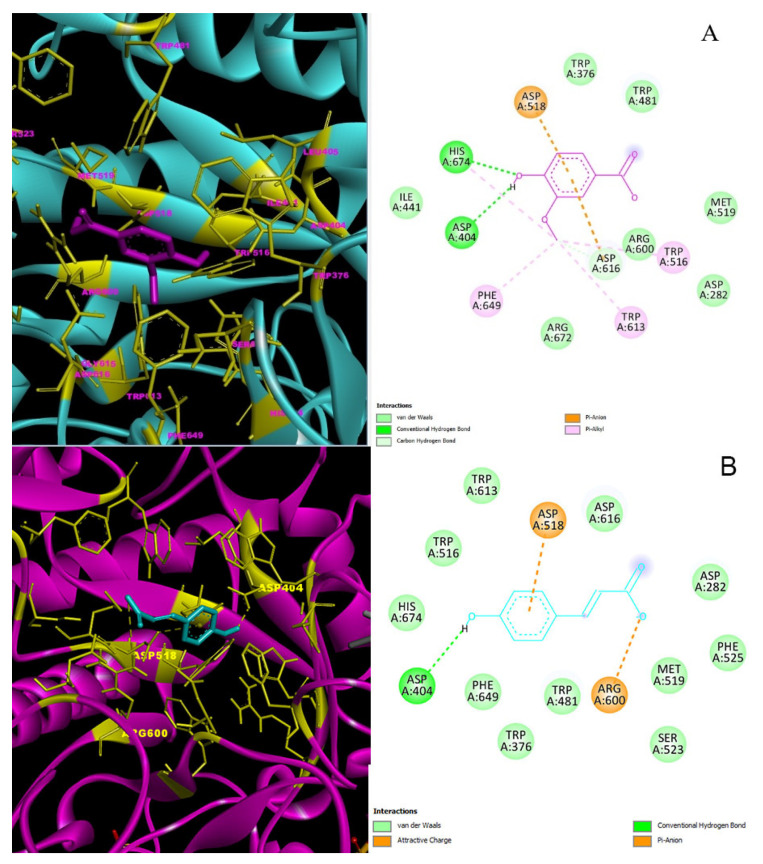
The 2D and 3D interaction profiles and best interaction poses of major phenolic compounds placed into the α-glycosidase (5NN8) by docking study: (**A**) Vanillic acid–α-glycosidase; (**B**) *p*-Coumaric acid–α-glycosidase.

**Figure 6 molecules-27-06907-f006:**
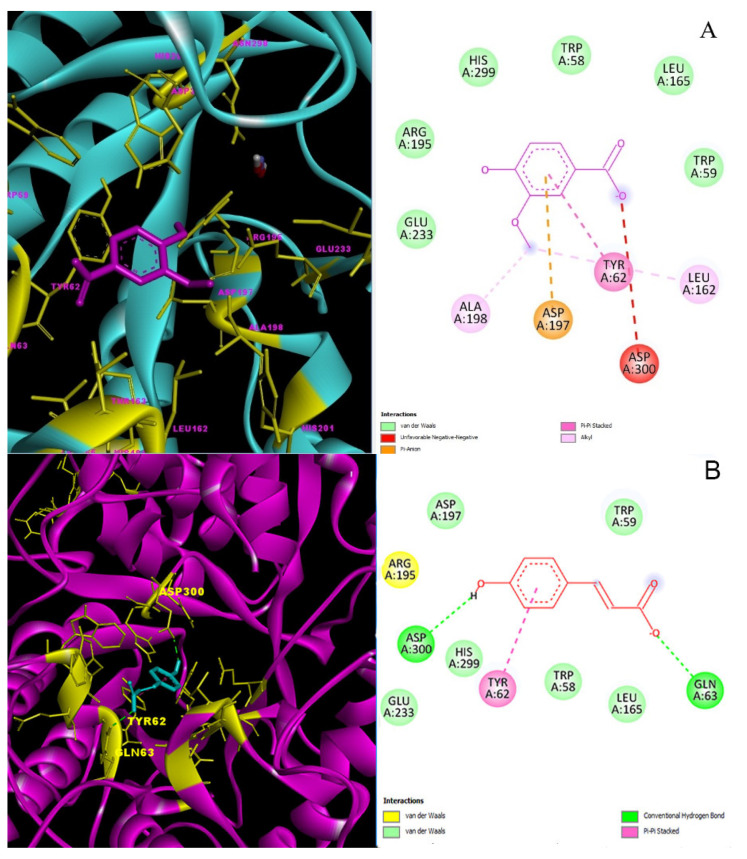
The 2D and 3D interaction profiles and best interaction poses of major phenolic compounds placed into the α-amylase enzyme (2QV4) by docking study: (**A**) Vanillic acid–α-amylase; (**B**) *p*-Coumaric acid–α-amylase.

**Table 1 molecules-27-06907-t001:** The half maximal concentrations (IC_50_; μg/mL) for ethanol extract of aerial parts (EDOA) and roots (EDOR) of sahlep (*D. osmanica*) and standards.

Compounds	DPPH Scavenging		ABTS Scavenging		Fe^2+^ Chelating	
	IC_50_	r^2^	IC_50_	r^2^	IC_50_	r^2^
**Ascorbic acid**	16.12 ± 0.003	0.9566	23.10 ± 0.001	0.9998	99.0 ± 0.036	0.9985
**α-Tocopherol**	23.10 ± 0.032	0.9825	15.40 ± 0.003	0.9866	330.0 ± 0.017	0.9109
**BHT**	31.50 ± 0.011	0.9754	26.65 ± 0.008	0.9717	14.75 ± 0.056	0.9646
**EDOA**	86.63 ± 0.010	0.9894	10.19 ± 0.002	0.9819	5.63 ± 0.033	0.9294
**EDOR**	115.50 ± 0.011	0.9794	19.80 ± 0.013	0.9564	46.20 ± 0.015	0.9000

**Table 2 molecules-27-06907-t002:** The reducing abilities of ethanol extract of aerial parts (EDOA) and roots (EDOR) of sahlep (*D. osmanica*) and standards.

Compounds	Fe^3+^ Reducing *		Cu^2+^ Reducing *		Fe^3+^-TPTZ Reducing *	
	λ_700_	r^2^	λ_450_	r^2^	λ_593_	r^2^
**Ascorbic acid (a)**	1.52 ± 0.028 ^b,c,d,e^	0.9970	1.07 ± 0.007 ^b,d,e^	0.9722	1.62 ± 0.015 ^b,d,e^	0.9930
**α-Tocopherol (b)**	0.99 ± 0.007 ^e^	0.9942	0.79 ± 0.061 ^d,e^	0.9986	0.76 ± 0.075 ^d,e^	0.9867
**BHT (c)**	1.27 ± 0.005 ^b,d,e^	0.9880	1.56 ± 0.089 ^a,b,d,e^	0.9978	0.91 ± 0.006 ^a,b,d,e^	0.9874
**EDOA (d)**	0.99 ± 0.003 ^b,e^	0.9553	0.67 ± 0.019	0.9747	0.52 ± 0.005	0.9722
**EDOR (e)**	0.83 ± 0.031	0.9423	0.72 ± 0.022 ^d^	0.9707	0.45 ± 0.006 ^d^	0.9673

* Different letters in the same column show significant difference between the means (*p* < 0.001 regarded as significant).

**Table 3 molecules-27-06907-t003:** The total phenolic (μgGAE/mL extract) and flavonoid (μgQE/mL extract) contents of ethanol extract of aerial parts (EDOA) and roots (EDOR) of sahlep (*D. osmanica*).

Extracts	Total Phenolics	Total Flavonoids
**EDOA**	12.73 ± 1.29	17.54 ± 1.85
**EDOR**	9.09 ± 0.64	3.28 ± 0.12

**Table 4 molecules-27-06907-t004:** The quantity of phenolics (mg/kg extract) in ethanol extract of *D. osmanica* aerial parts (EDOA) and roots (EDOR) determined by LC-HRMS.

Compounds	EDOA	EDOR
Ascorbic acid	30.62	42.93
Chlorogenic acid	7.31	3.21
Fumaric acid	1542.92	822.95
Verbascoside	0.95	1.61
Orientin	0.15	5.78
Caffeic acid	14.72	13.16
(+)-t*rans* taxifolin	<LOD	<LOD
Luteolin-7-rutinoside	0.53	<LOD
Vanillic acid	62.22	16.56
Naringin	<LOD	0.00
Luteolin 7-glucoside	<LOD	1.02
*p*-Coumaric acid	541.49	559.22
Hesperidin	0.29	0.39
Rutine	3.87	2.97
Rosmarinic acid	3.08	21.11
Hyperoside	18.26	16.05
Dihydrokaempferol	0.44	0.20
Apigenin 7-glucoside	<LOD	0.03
Quercitrin	29.31	6.05
Myricetin	<LOD	<LOD
Quercetin	0.95	1.11
Salicylic acid	<LOD	<LOD
Naringenin	8.63	3.32
Luteolin	1.11	1.41
Nepetin	0.69	<LOD
Apigenin	0.71	0.35
Hispidulin	4.49	4.46
Isosakuranetin	<LOD	<LOD
Penduletin	<LOD	0.59
Caffeic acid phenethyl ester	1.77	0.15
Chrysin	5.57	1.26
Acacetin	4.59	<LOD
Emodin	<LOD	0.02

**Table 5 molecules-27-06907-t005:** The enzyme inhibition (IC_50_, μg/mL) of EDOA and EDOR against α-amylase, α-glycosidase, and acetylcholinesterase.

Enzymes	EDOA		EDOR		Standards
	IC_50_	r^2^	IC_50_	r^2^	IC_50_
**α-Glycosidase ^a^**	1.098	0.9545	0.442	0.9498	22.80
**α-Amylase ^a^**	0.726	0.9860	0.415	0.9747	10.01
**Acetylcholinesterase ^b^**	1.809	0.9722	2.466	0.9826	0.124

^a^ Acarbose (ACR) had been used as positive inhibitor for α-glycosidase and α-amylase and taken from references [21,23], respectively. ^b^ Tacrine was used as positive control for AChE and taken from reference [24].

**Table 6 molecules-27-06907-t006:** Molecular interactions of the AChE, α-amylase, and α-glycosidase with the major phenolic compounds of *Dactylorhiza osmanica* (vanillic acid, *p*-coumaric acid, and quercitrin).

Complex	Docking Scores (kcal/mol)	Types of Interactions	Interacting Residues
**AChE** **(4EY7)–Vanillic acid**	−6.8	H-bonding π-π stacked π alkyl	Tyr-133, Gly-121 Trp-86 Tyr-124, Trp-86
**AChE** **(4EY7)–Quercitrin**	−8.8	H-bonding C-H bonding π-π stacked	Gln-291, Arg-296, Glu-292, Trp-286
**α-Glycosidase (5NN8)–Vanillic acid**	−5.6	H-bonding π alkyl π anion	Asp-616, Asp-404, His-674 Trp-516, Trp-613, Phe-649, His-674, Asp-518
**α-Glycosidase (5NN8)–*p*-Coumaric acid**	−6.5	H-bonding π anion	Asp-404, Asp-518, Arg-600
**α-A** **mylase (2QV4)–Vanillic acid**	−5.6	H-bonding π-π stacked, π anion π alkyl	Arg-195, His-299 Tyr-62, Asp-197 Ala-198, Leu-162
**α-A** **mylase (2QV4** **)–*p*-Coumaric acid**	−5.6	H-bonding π-π stacked	Asp-300, Gln-63 Tyr-62

**Table 7 molecules-27-06907-t007:** Validation and uncertainty parameters for phenolic compounds.

Compound	Molecular Formula	m/z	Ionization Mode	Linear Range	Linear Regression Equation	LOD/LOQ	R²
Ascorbic acid	C_6_H_8_O_6_	175.0248	Negative	0.5–10	y = 0.00347x − 0.00137	0.39/1.29	0.9988
Chlorogenic acid	C_16_H_18_O_9_	353.0878	Negative	0.05–10	y = 0.00817x + 0.000163	0.02/0.06	0.9994
Fumaric acid	C_4_H_4_O_4_	115.0037	Negative	0.1–10	y = 0.00061x − 0.0000329	0.05/0.17	0.9991
Verbascoside	C_29_H_36_O_15_	623.1981	Negative	0.1–10	y = 0.00758x + 0.000563	0.03/0.1	0.9995
Orientin	C_21_H_20_O_11_	447.0933	Negative	0.1–10	y = 0.00757x + 0.000347	0.01/0.03	0.999
Caffeic acid	C_9_H_8_O_4_	179.0350	Negative	0.3–10	y = 0.0304x + 0.00366	0.08/0.27	0.9993
(+)-t*rans* taxifolin	C_15_H_12_O_7_	303.0510	Negative	0.3–10	y = 0.0289x + 0.00537	0.01/0.03	0.9978
Luteolin-7-rutinoside	C_27_H_30_O_15_	593.1512	Negative	0.1–10	y = 0.00879x + 0.000739	0.01/0.03	0.9988
Vanillic acid	C_8_H_8_O_4_	167.0350	Negative	0.3–10	y = 0.00133x + 0.0003456	0.1/0.33	0.9997
Naringin	C_27_H_32_O_14_	579.1719	Negative	0.05–10	y = 0.00576x − 0.000284	0.01/0.03	0.999
Luteolin 7-glycoside	C_21_H_20_O_11_	447.0933	Negative	0.1–7	y = 0.0162x + 0.00226	0.01/0.03	0.9961
*p*-Coumaric acid	C_9_H_8_O_3_	163.0401	Negative	1 + 10	y = 0.000324x − 0.0000641	0.32/1.02	0.9988
Hesperidin	C_28_H_34_O_15_	609.1825	Negative	0.05–10	y = 0.00423x + 0.0000138	0.01/0.03	0.999
Rutin	C_27_H_30_O_16_	609.1461	Negative	0.05–10	y = 0.00329x − 0.00005576	0.01/0.03	0.999
Rosmarinic acid	C_18_H_16_O_8_	359.0772	Negative	0.05–10	y = 0.00717x − 0.0003067	0.01/0.03	0.999
Hyperoside	C_21_H_20_O_12_	463.0882	Negative	0.05–10	y = 0.0072x − 0.00003096	0.01/0.03	1.000
Dihydrokaempferol	C_15_H_12_O_6_	287.0561	Negative	0.3–7	y = 0.0756x + 0.0118	0.01/0.03	0.995
Apigenin 7-glucoside	C_21_H_20_O_10_	431.0984	Negative	0.3–7	y = 0.0246x + 0.00306	0.01/0.03	0.996
Quercitrin	C_21_H_20_O_11_	447.0933	Negative	0.05–10	y = 0.0179 + 0.0003331	0.01/0.03	0.999
Myricetin	C_15_H_10_O_8_	317.0303	Negative	0.1–10	y = 0.0202x + 0.00165	0.01/0.03	0.9993
Quercetin	C_15_H_10_O_7_	301.0354	Negative	0.1–10	y = 0.0509x + 0.00467	0.01/0.03	0.9978
Salicylic acid	C_7_H_6_O_3_	137.0244	Negative	0.3–10	y = 0.0361x + 0.00245	0.01/0.03	0.9982
Naringenin	C_15_H_12_O_5_	271.0612	Negative	0.1–10	y = 0.0281x + 0.00182	0.01/0.03	0.9995
Luteolin	C_15_H_10_O_6_	285.0405	Negative	0.1–10	y = 0.117x + 0.00848	0.01/0.03	0.998
Nepetin	C_16_H_12_O7	315.0510	Negative	0.05–10	y = 0.0853x + 0.00269	0.01/0.03	0.9992
Apigenin	C_15_H_10_O_5_	269.0456	Negative	0.3–10	y = 0.104x + 0.0199	0.01/0.03	0.9998
Hispidulin	C_16_H_12_O_6_	301.0707	Positive	0.05–10	y = 0.02614x + 0.0003114	0.01/0.03	0.9993
Isosakuranetin	C_16_H_14_O_5_	285.0769	Negative	0.05–10	y = 0.0235x + 0.000561	0.01/0.03	0.999
Penduletin	C_18_H_16_O_7_	343.0823	Negative	0.3–10	y = 0.0258x + 0.00253	0.01/0.03	0.999

## Data Availability

Data are provided in a publicly accessible repository.

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
