# Peer review of "Sahlep (Dactylorhiza osmanica): Phytochemical Analyses by LC-HRMS, Molecular Docking, Antioxidant Activity, and Enzyme Inhibition Profiles"

_molecules, 2022, doi:10.3390/molecules27206907_

Round 1
Reviewer 1 Report
The authors aim was to evaluate the antioxidant capacities of Sahlep using different methods. The study objective is clear, the applied methods are satisfactory. The most valuable part of the manuscript is the 2D and 3D figures about the possible interaction profiles of major phenolic compounds. The authors supported their works by 88 references.
Overall: the study objective must be particularly determined in the end of introduction part. The figures qualities must be improved. There is no SD values in the figures. More figure legends are also needed including the short explanation and the statistical evaluations.
Intensive English correction is needed including the elimination of spelling mistakes.
Author Response
RESPONSES TO REVIEWER-1
The authors aim was to evaluate the antioxidant capacities of Sahlep using different methods. The study objective is clear; the applied methods are satisfactory. The most valuable part of the manuscript is the 2D and 3D figures about the possible interaction profiles of major phenolic compounds. The authors supported their works by 88 references.
Overall: the study objective must be particularly determined in the end of introduction part. The figures qualities must be improved. There is no SD values in the figures. More figure legends are also needed including the short explanation and the statistical evaluations.
RESPONSE:
Intensive English correction is needed including the elimination of spelling mistakes.
RESPONSE: Many thanks to the reviewer due to his/her positive opinion.
Reviewer 2 Report
The article presented to me for review entitled: "Sahlep (Dactylorhiza osmanica): Phytochemical Analyzes by LC-HRMS, Molecular Docking, Antioxidant Activity and Enzyme Inhibition Profiles" presents an interesting topic worth publishing in Molecules. The Authors used advanced instrumental methods to assess the biological potential of Dactylorhiza osmanica. I believe that the research is rightly planned and the manuscript is properly structured.
Before publishing, please correct the following points:
1 / keywords cannot duplicate with the topic; their task is to increase the visibility of the article
2 / tables - please leave only the horizontal outer edges and inner edges only in the headers - to make them more aesthetic
3 / Charts - please remove the borders; standardize the size and type of font; Please leave graph captions in the caption only, remove from the graph
4 / Table 4 - please present the LOD, LOQ, equations and curve fitting for the determined compounds
5 / Section 4. Materials and Methods
- please provide all the reagents and standards used in the tests, together with their source of origin; information is currently incomplete
- for each device used in the tests, please provide the following record: name (model, manufacturer, city, country)
- methodologies: please provide FULL methodological descriptions (including detailed chromatographic separation data) for all performed analyzes; when specifying the concentration%, please write (m/m, m/v, etc.); if a standard curve was used for the calculations, please provide the equation and the fitting as well as the concentration range used for its preparation;
6 / Please insert a chromatogram from the HPLC analysis - it is worth showing the chromatogram from the analysis of samples and standards
7 / References - please adjust to Molecules requirements; add DOI numbers
Author Response
RESPONSES TO REVIEWER-2
The article presented to me for review entitled: "Sahlep (Dactylorhiza osmanica): Phytochemical Analyzes by LC-HRMS, Molecular Docking, Antioxidant Activity and Enzyme Inhibition Profiles" presents an interesting topic worth publishing in Molecules. The Authors used advanced instrumental methods to assess the biological potential of Dactylorhiza osmanica. I believe that the research is rightly planned and the manuscript is properly structured.
Before publishing, please correct the following points:
RESPONSE: Many thanks to the reviewer due to his/her opinion.
1 / keywords cannot duplicate with the topic; their task is to increase the visibility of the article
RESPONSE: The keywords given for the article were meticulously chosen to increase the visibility of the article and reflect the basic elements of the subject.
2 / tables - please leave only the horizontal outer edges and inner edges only in the headers - to make them more aesthetic
RESPONSE: Drawing of tables has been corrected using Template, which given in Instructions for authors.
3 / Charts - please remove the borders; standardize the size and type of font; Please leave graph captions in the caption only, remove from the graph
RESPONSE: We've removed border lines in charts, standardized font size and type, removed chart descriptions from charts, and given them in the title only.
4 / Table 4 - please present the LOD, LOQ, equations and curve fitting for the determined compounds
RESPONSE: The requested information was added as section “4.8.3. Method Validation” and related references and Table 7 were provided.
5 / Section 4. Materials and Methods
- please provide all the reagents and standards used in the tests, together with their source of origin; information is currently incomplete
RESPONSE: All the reagents and standards together with their source of origin; information used in the tests were provided.
- for each device used in the tests, please provide the following record: name (model, manufacturer, city, country)
RESPONSE: The name of model, manufacturer, city, country for each device used in the tests were supplied.
- methodologies: please provide FULL methodological descriptions (including detailed chromatographic separation data) for all performed analyzes; when specifying the concentration%, please write (m/m, m/v, etc.); if a standard curve was used for the calculations, please provide the equation and the fitting as well as the concentration range used for its preparation;
RESPONSE: The requested additions were given in the text as following:
“4.8.1. Preparation of samples and conditions for LC-HRMS Analysis
The phenolic contents in EDOA and EDOR were determined by LC-HRMS analysis [79,80]. LC-HRMS experiments were achieved on a Thermo ORBITRAP Q-EXACTIVE mass spectrometry (Bremen, Germany) equipped with a Troyasil (Istanbul Turkiye) C18 column (150 x 3 mm i.d., 3 µm particle size) for measurements. The mobile phases A and B were composed of 1% formic acid-water and 1% formic acid-methanol, respectively. While the gradient programme of which was 0-1.00 min 50% A and 50% B, 1.01-6.00 min 100% B, and finally 6.01-10 min 50% A and 50% B.”
“4.8.2. LC-HRMS procedure and optimization of HPLC methods
The finest mobile phase included an acidified methyl alcohol and water gradient by HPLC method [82]. The identification of the phenolics was made by comparing the retention times of the standard phenolics (in the range of purity 95-99% see section chemicals) and HRMS data of ILMER in Bezmialem Vakıf University, Dihydrocapsaicin (95% purity) used as internal standard (IS) for LC-HRMS for reducing of repeatability of caused by external effects like ionization repeatability in mass spectrometry measurements. 0.1 g/L dihydrocapsaicin (97%, Sigma-Aldrich) solution was used as an IS. Linear range of the standard solutions were given as mg/kg in Table S1. The mass parameters related to target compounds is summarized in the Table 4 [81-83].”
“4.8.3. Method Validation
Validation of the LC-HRMS method was performed using analytical standards of corresponding compounds (see section 4.1) with using the target ions (Table S1). Considering the EURACHEM/CITAC Guide [84] and purpose of the method, the validation parameters were selected as linearity, recovery, repeatability, LOD and LOQ for the applied method. The limit of detection (LODs) of the method for each compound was determined according to the following equation: LOD or LOQ = κSDa/b, where 3 for LOQ and κ= 3 for LOD, SDa represents the standard deviation of the intercept, and b represents the slope (Table S1) [82,85]”
6 / Please insert a chromatogram from the HPLC analysis - it is worth showing the chromatogram from the analysis of samples and standards
RESPONSE: A chromatogram from the HPLC analysis for samples and standards was given as Figure 3. All figures were renumbered again.
7 / References - please adjust to Molecules requirements; add DOI numbers
RESPONSE: doi numbers of all articles were given.
Reviewer 3 Report
The manuscript by Kiziltas et al described the study of Sahlep extracts in antioxidation, and enzyme inhibition applications. Both the aerial and root extracts (EDOA and EDOR) had some extent of antioxidant activities. It was also found that p-coumaric acid and fumaric acid are the main polyphenols in EDOA and EDOR. The authors also tested inhibition of α-glucosidase, α-amylase and AChE with EDOA and EDOR, and performed docking experiments which showed potential binding.
The data showing antioxidant activities and enzyme inhibition is convincing, but the docking results can only be used for reference. Actually, p-coumaric acid has been showed to have no inhibition toward AChE. (doi: 10.1016/S2222-1808(13)60088-2) The manuscript tried to connect Sahlep extracts to several functions and diseases, but it is unclear how the extracts would work in disease relieve.
In summary, despite the considerable amount of data included in this paper, this reviewer cannot recommend this paper for publication.
Author Response
The manuscript by Kiziltas et al described the study of Sahlep extracts in antioxidation, and enzyme inhibition applications. Both the aerial and root extracts (EDOA and EDOR) had some extent of antioxidant activities. It was also found that p-coumaric acid and fumaric acid are the main polyphenols in EDOA and EDOR. The authors also tested inhibition of α-glucosidase, α-amylase and AChE with EDOA and EDOR, and performed docking experiments which showed potential binding.
The data showing antioxidant activities and enzyme inhibition is convincing, but the docking results can only be used for reference. Actually, p-coumaric acid has been showed to have no inhibition toward AChE. (doi: 10.1016/S2222-1808(13)60088-2) The manuscript tried to connect Sahlep extracts to several functions and diseases, but it is unclear how the extracts would work in disease relieve.
RESPONSE: We excluded the molecular docking interactions of p-coumaric acid, which is the most abundant phenolic acid in the extracts, and the AChE enzyme, as suggested by reviewer. Instead, we studied the molecular docking interactions of Vanillic acid as the second most abundant phenolic compound in the extract, and the AChE enzyme. We examined and emphasized the inhibitory effects of Sahlep extracts on acetylcholinesterase (AChE), α-glucosidase and α-amylase enzymes in several parts of the article. In particular, it is known that the inhibition of these three enzymes has therapeutic effects against Alzheimer's disease and diabetes. Also, the inhibition of these enzymes is one of the most common clinical treatments for both diseases. As stated by rewiver p-coumaric acid has been showed to have no inhibition toward AChE (doi: 10.1016/S2222-1808(13)60088-2). The interaction of p-coumaric acid and AChE was replaced with quercitrin and AChE due to quercitrin is third most plentiful phenolic compound in EDOA and EDOR
-In summary, despite the considerable amount of data included in this paper, this reviewer cannot recommend this paper for publication.
RESPONSE: We made the corrections suggested by all the reviewers. I hope that after these corrections, the referee will change his decision and express his positive opinion in favor of the acceptance of the manuscript.
Round 2
Reviewer 2 Report
I recommend this version for publication.
Reviewer 3 Report
The authors addressed my concerns.